# Doubly Robust Counterfactual Classification

**Kwangho Kim**
Harvard Medical School
kkim@hcp.med.harvard.edu

**Edward H. Kennedy**
Carnegie Mellon University
edward@stat.cmu.edu

**José R. Zubizarreta**
Harvard University
zubizarreta@hcp.med.harvard.edu

## Abstract

We study counterfactual classification as a new tool for decision-making under hypothetical (contrary to fact) scenarios. We propose a doubly-robust nonparametric estimator for a general counterfactual classifier, where we can incorporate flexible constraints by casting the classification problem as a nonlinear mathematical program involving counterfactuals. We go on to analyze the rates of convergence of the estimator and provide a closed-form expression for its asymptotic distribution. Our analysis shows that the proposed estimator is robust against nuisance model misspecification, and can attain fast $\sqrt{n}$ rates with tractable inference even when using nonparametric machine learning approaches. We study the empirical performance of our methods by simulation and apply them for recidivism risk prediction.

## 1 Introduction

*Counterfactual or potential outcomes* are often used to describe how an individual would respond to a specific treatment or event, irrespective of whether the event actually takes place. Counterfactual outcomes are commonly used for causal inference, where we are interested in measuring the effect of a treatment on an outcome variable [15, 16, 45].

Recently, counterfactual outcomes have also proved useful for predicting outcomes under hypothetical interventions. This is commonly referred to as *counterfactual prediction*. Counterfactual prediction can be particularly useful to inform decision-making in clinical practice. For example, in order for physicians to make effective treatment decisions, they often need to predict risk scores assuming no treatment is given; if a patient's risk is relatively low, then she or he may not need treatment. However, when a treatment is initiated after baseline, simply operationalizing the hypothetical treatment as another baseline predictor will rarely give the correct (counterfactual) risk estimates because of confounding [58]. Counterfactual prediction can be also helpful when we want our prediction model developed in one setting to yield predictions successfully transportable to other settings with different treatment patterns. Suppose that we develop our risk prediction model in a setting where most patients have access to an effective (post-baseline) treatment. However, if we deploy our factual prediction model in a new setting in which few individuals have access to the treatment, our model is likely to fail in the sense that it may not be able to accurately identify high-risk individuals. Counterfactual prediction may allow us to achieve more robust model performance compared to factual prediction, even when model deployment influences behaviors that affect risk. [see, e.g., 10, 27, 54, for more examples].

However, the problem of counterfactual prediction brings challenges that do not arise in typical prediction problems because the data needed to build the predictive models are inherently not fully

36th Conference on Neural Information Processing Systems (NeurIPS 2022).

observable. Surprisingly, while the development of modern prediction modeling has greatly enriched the counterfactual-outcome-based causal inference particularly via semi-parametric methods [20, 23], the use of causal inference to improve prediction modeling has received less attention [see, e.g., 10, 46, for a discussion on the subject].

In this work, we study counterfactual classification, a special case of counterfactual prediction where the outcome is discrete. Our approach allows investigators to flexibly incorporate various constraints into the models, not only to enhance their predictive performance but also to accommodate a wide range of practical constraints relevant to their classification tasks. Counterfactual classification poses both theoretical and practical challenges, as a result of the fact that in our setting, even without any constraints, the estimand is not expressible as a closed form functional unlike typical causal inference problems. We tackle this problem by framing counterfactual classification as nonlinear stochastic programming with counterfactual components.

## 1.1 Related Work

Our work lies at the intersection of causal inference and stochastic optimization.

Counterfactual prediction is closely related to estimation of the conditional average treatment effect (CATE) in causal inference, which plays a crucial role in precision medicine and individualized policy. Let $Y^a$ denote the counterfactual outcome that would have been observed under treatment or intervention $A = a$, $A \in \{0, 1\}$. The CATE for subjects with covariate $X = x$ is defined as $\tau(x) = \mathbb{E}[Y^1 - Y^0 \mid X = x]$. There exists a vast literature on estimating CATE. These include some important early works assuming that $\tau(x)$ follows some known parametric form [e.g., 44, 52, 55]. But more recently, there has been an effort to leverage flexible nonparametric machine learning methods [e.g., 1, 3, 22, 25, 29, 31, 39, 57]. A desirable property commonly held in the above CATE estimation methods is that the function $\tau(x)$ may be more structured and simple than its component main effect function $\mathbb{E}[Y^a \mid X = x]$.

In counterfactual prediction, however, we are fundamentally interested in predicting $Y^a$ conditional on $X = x$ under a "single" hypothetical intervention $A = a$, as opposed to the contrast of the conditional mean outcomes under two (or more) interventions as in CATE. Counterfactual prediction is often useful to support decision-making on its own. There are settings where estimating the contrast effect or relative risk is less relevant than understanding what may happen if a subject was given a certain intervention. As mentioned previously, this is particularly the case in clinical research when predicting risk in relation to treatment started after baseline [10, 27, 46, 54]. Moreover, in the context of multi-valued treatments, it can be more useful to estimate each individual conditional mean potential outcome separately than to estimate all the possible combinations of relative effects.

With no constraints, under appropriate identification assumptions (e.g., (C1)-(C3) in Section 2), counterfactual prediction is equivalent to estimating a standard regression function $\mathbb{E}[Y \mid X, A = a]$ so in principle one could use any regression estimator. This direct modeling or *plug-in* approach has been used for counterfactual prediction in randomized controlled trials [e.g., 26, 38] or as a component of CATE estimation methods [e.g., 3, 29]. An issue arises when we are estimating a projection of this function onto a finite-dimensional model, or where we instead want to estimate $\mathbb{E}[Y^a \mid V] = \mathbb{E}\{\mathbb{E}[Y \mid X, A = a] \mid V\}$ for some smaller subset $V \subset X$ (e.g., under runtime confounding [9]), which typically renders the plug-in approach suboptimal. Moreover, the resulting estimator fails to have double robustness, a highly desirable property which provides an additional layer of robustness against model misspecification [4].

On the other hand, we often want to incorporate various constraints into our predictive models. Such constraints are often used for flexible penalization [18] or supplying prior information [13] to enhance model performance and interpretability. They can also be used to mitigate algorithmic biases [6, 14]. Further, depending on the scientific question, practitioners occasionally have some constraints which they wish to place on their prediction tasks, such as targeting specific sub-populations, restricting sign or magnitude on certain regression coefficients to be consistent with common sense, or accounting for the compositional nature of the data [7, 19, 28]. In the plug-in approach, however, it is not clear how to incorporate the given constraints into the modeling process.

In our approach, we directly formulate and solve an optimization problem that minimizes counterfactual classification risk, where we can flexibly incorporate various forms of constraints. Optimization problems involving counterfactuals or *counterfactual optimization* have not been extensively studied,

with few exceptions [e.g., 24, 30, 33, 34]. Our results are closest to [33] and [24], which study counterfactual optimization in a class of quadratic and nonlinear programming problems, respectively, yet this approach i) is not applicable to classification where the risk is defined with respect to the cross-entropy, and ii) considers only linear constraints.

As in [24], we tackle the problem of counterfactual classification from the perspective of stochastic programming. The two most common approaches in stochastic programming are stochastic approximation (SA) and sample average approximation (SAA) [e.g., 36, 50]. However, since i) we cannot compute sample moments or stochastic subgradients that involve unobserved counterfactuals, and ii) the SA and SAA approaches cannot harness efficient estimators for counterfactual components, e.g., doubly-robust or semiparametric estimators with cross-fitting [8, 37], more general approaches beyond the standard SA and SAA settings should be considered [e.g., 47–49] at the expense of stronger assumptions on the behavior of the optimal solution and its estimator.

## 1.2 Contribution

We study counterfactual classification as a new decision-making tool under hypothetical (contrary to fact) scenarios. Based on semiparametric theory for causal inference, we propose a doubly-robust, nonparametric estimator that can incorporate flexible constraints into the modeling process. Then we go on to analyze rates of convergence and provide a closed-form expression for the asymptotic distribution of our estimator. Our analysis shows that the proposed estimator can attain fast $\sqrt{n}$ rates even when its nuisance components are estimated using nonparametric machine learning tools at slower rates. We study the finite-sample performance of our estimator via simulation and provide a case based on real data. Importantly, our algorithm and analysis are applicable to other problems in which the estimand is given by the solutions to a general nonlinear optimization problem whose objective function involves counterfactuals, where closed-form solutions are not available.

## 2 Problem and Setup

Suppose that we have access to an i.i.d. sample $(Z_1, ..., Z_n)$ of $n$ tuples $Z = (Y, A, X) \sim \mathbb{P}$ for some distribution $\mathbb{P}$, binary outcome $Y \in \{0, 1\}$, covariates $X \in \mathcal{X} \subset \mathbb{R}^{d_x}$, and binary intervention $A \in \mathcal{A} = \{0, 1\}$. For simplicity, we assume $A$ and $Y$ are binary, but in principle they can be multi-valued. We consider a general setting where only a subset of covariates $V \subseteq X$ can be used for predicting the counterfactual outcome $Y^a$. This allows for *runtime confounding*, where factors used by decision-makers are recorded in the training data but are not available for prediction (see [9] and references therein). We are concerned with the following constrained optimization problem

$$\begin{aligned} &\underset{\beta \in \mathcal{B}}{\text{minimize}} \quad \mathcal{L}\left(Y^a, \sigma(\beta, \boldsymbol{b}(V))\right) := -\mathbb{E}\left\{Y^a \log \sigma(\beta, \boldsymbol{b}(V)) + (1 - Y^a) \log(1 - \sigma(\beta, \boldsymbol{b}(V)))\right\} \\ &\text{subject to} \quad \beta \in \mathcal{S} := \{\beta \mid g_j(\beta) \le 0, j \in J\} \end{aligned}$$

(P)

for some compact subset $\mathcal{B} \in \mathbb{R}^k$, known $C^2$-functions $g_j : \mathcal{B} \to \mathbb{R}$, $\sigma : \mathcal{B} \times \mathbb{R}^{k'} \to (0, 1)$, and the index set $J = \{1, ..., m\}$ for the inequality constraints. Here, $\sigma$ is the score function and $\boldsymbol{b}(V) = [b_1(V), ..., b_{k'}(V)]^\top$ represents a set of basis functions for $V$ (e.g., truncated power series, kernel or spline basis functions, etc.). Note that we do not need to have $k = k'$; for example, depending on the modeling techniques, it is possible to have a much larger number of model parameters than the number of basis functions, i.e., $k > k'$. $\mathcal{L}(Y^a, \sigma(\beta, \boldsymbol{b}(V)))$ is our classification risk based on the cross-entropy. $\mathcal{S}$ consists of deterministic inequality constraints[1] and can be used to pursue a variety of practical purposes described in Section 1. Let $\beta^*$ denote an optimal solution in (P). $\beta^*$ is our optimal model parameters (coefficients) that minimize the counterfactual classification risk under the given constraints.

**Classification risk and score function.** Our classification risk $\mathcal{L}(Y^a, \sigma(\beta, \boldsymbol{b}(V)))$ is defined by the expected cross entropy loss between $Y^a$ and $\sigma(\beta, \boldsymbol{b}(V))$. In order to estimate $\beta^*$, we first need to estimate this classification risk. Since it involves counterfactuals, the classification risk cannot be identified from observed data unless certain assumptions hold, which will be discussed shortly. The form of the score function $\sigma(\beta, \boldsymbol{b}(V))$ depends on the specific classification technique we are using. Our default choice for $\sigma$ is the sigmoid function with $k = k'$, which makes the classification

---

[1]Equality constraint can be always expressed by a pair of inequality constraints.

risk strictly convex with respect to $\beta$. It should be noted, however, that more complex and flexible classification techniques (e.g., neural networks) can also be used without affecting the subsequent results, as long as they satisfy the required regularity assumptions discussed later in Section 4. Importantly, our approach is nonparametric; $\beta^*$ is the parameter of the best linear classifier with the sigmoid score in the expanded feature space spanned by $\boldsymbol{b}(V)$, but we never assume an exact 'log-linear' relationship between $Y^a$ and $\boldsymbol{b}(V)$ as in ordinary logistic regression models.

**Identification.** To estimate the counterfactual quantity $\mathcal{L}(Y^a, \sigma(\beta, \boldsymbol{b}(V)))$ from the observed sample $(Z_1, ..., Z_n)$, it must be expressed in terms of the observational data distribution $\mathbb{P}$. This can be accomplished via the following standard causal assumptions [e.g., 17, Chapter 12]:

- (C1) *Consistency*: $Y = Y^a$ if $A = a$
- (C2) *No unmeasured confounding*: $A \perp\!\!\!\perp Y^a \mid X$
- (C3) *Positivity*: $\mathbb{P}(A = a|X) > \varepsilon$ a.s. for some $\varepsilon > 0$

(C1) - (C3) will be assumed throughout this paper. Under these assumptions, our classification risk is identified as

$$\mathcal{L}(\beta) = -\mathbb{E}\left\{\mathbb{E}\left[Y \mid X, A = a\right] \log \sigma(\beta, \boldsymbol{b}(V)) + (1 - \mathbb{E}\left[Y \mid X, A = a\right]) \log(1 - \sigma(\beta, \boldsymbol{b}(V)))\right\}, \tag{1}$$

where we let $\mathcal{L}(\beta) \equiv \mathcal{L}(Y^a, \sigma(\beta, \boldsymbol{b}(V)))$. Since we use the sigmoid function with an equal number of model parameters as basis functions, for clarity, hereafter we write $\sigma(\beta^\top \boldsymbol{b}(V)) = \sigma(\beta, \boldsymbol{b}(V))$. It is worth noting that even though we develop the estimator under the above set of causal assumptions, one may extend our methods to other identification strategies and settings (e.g., those of instrumental variables and mediation), since our approach is based on the analysis of a stochastic programming problem with generic estimated objective functions (see Appendix B).

**Notation.** Here we specify the basic notation used throughout the paper. For a real-valued vector $v$, let $\|v\|_2$ denote its Euclidean or $L_2$-norm. Let $\mathbb{P}_n$ denote the empirical measure over $(Z_1, ..., Z_n)$. Given a sample operator $h$ (e.g., an estimated function), let $\mathbb{P}$ denote the conditional expectation over a new independent observation $Z$, as in $\mathbb{P}(h) = \mathbb{P}\{h(Z)\} = \int h(z)d\mathbb{P}(z)$. Use $\|h\|_{2,\mathbb{P}}$ to denote the $L_2(\mathbb{P})$ norm of $h$, defined by $\|h\|_{2,\mathbb{P}} = \left[\mathbb{P}(h^2)\right]^{\frac{1}{2}} = \left[\int h(z)^2 d\mathbb{P}(z)\right]^{\frac{1}{2}}$. Finally, let $\mathsf{s}^*(P)$ denote the set of optimal solutions of an optimization program $P$, i.e., $\beta^* \in \mathsf{s}^*(P)$, and define $\mathsf{dist}(x, S) = \inf\left\{\|x - y\|_2 : y \in S\right\}$ to denote the distance from a point $x$ to a set $S$.

## 3 Estimation Algorithm

Since (P) is not directly solvable, we need to find an approximating program of the "true" program (P). To this end, we shall first discuss the problem of obtaining estimates for the identified classification risk (1). To simplify notation, we first introduce the following nuisance functions

$$\pi_a(X) = \mathbb{P}[A = a \mid X],$$
$$\mu_a(X) = \mathbb{E}[Y \mid X, A = a],$$

and let $\widehat{\pi}_a$ and $\widehat{\mu}_a$ be their corresponding estimators. $\pi_a$ and $\mu_a$ are referred to as the propensity score and outcome regression function, respectively.

A natural estimator for (1) is given by

$$\widehat{\mathcal{L}}(\beta) = -\mathbb{P}_n\left\{\widehat{\mu}_a(X) \log \sigma(\beta^\top \boldsymbol{b}(V)) + (1 - \widehat{\mu}_a(X)) \log(1 - \sigma(\beta^\top \boldsymbol{b}(V)))\right\}, \tag{2}$$

where we simply plug in the regression estimates $\widehat{\mu}_a$ into the empirical average of (1). Here, we construct a more efficient estimator based on the semiparametric approach in causal inference [21, 23]. Let

$$\varphi_a(Z; \eta) = \frac{\mathbb{1}(A = a)}{\pi_a(X)}\{Y - \mu_A(X)\} + \mu_a(X),$$

denote the uncentered efficient influence function for the parameter $\mathbb{E}\{\mathbb{E}[Y \mid X, A = a]\}$, where nuisance functions are defined by $\eta = \{\pi_a(X), \mu_a(X)\}$. Then it can be deduced that for an arbitrary

---

**Algorithm 1:** Doubly robust estimator for counterfactual classification

---

1 **input:** $\boldsymbol{b}(\cdot), K$
2 Draw $(B_1, ..., B_n)$ with $B_i \in \{1, ..., K\}$
3 **for** $b = 1, ..., K$ **do**
4      Let $D_0 = \{Z_i : B_i \neq b\}$ and $D_1 = \{Z_i : B_i = b\}$
5      Obtain $\widehat{\eta}_{-b}$ by constructing $\widehat{\pi}_a, \widehat{\mu}_a$ on $D_0$
6      $M_{1,b}(\beta) \leftarrow$ empirical average of $\varphi_a(Z; \widehat{\eta}_{-b}) \log \sigma(\beta^\top \boldsymbol{b}(V))$ over $D_1$
7      $M_{0,b}(\beta) \leftarrow$ empirical average of $(1 - \varphi_a(Z; \widehat{\eta}_{-b})) \log(1 - \sigma(\beta^\top \boldsymbol{b}(V)))$ over $D_1$
8 $\widehat{\mathcal{L}}(\beta) \leftarrow \sum_{b=1}^{K} \left\{ \frac{1}{n} \sum_{i=1}^{n} \mathbb{1}(B_i = b) \right\} (M_{1,b}(\beta) + M_{0,b}(\beta))$
9 **solve** ($\widehat{\mathsf{P}}$) with $\widehat{\mathcal{L}}(\beta)$

---

fixed real-valued function $h : \mathcal{X} \to \mathbb{R}$, the uncentered efficient influence function for the parameter $\psi_a := \mathbb{E}\{\mathbb{E}[Y \mid X, A = a]h(X)\}$ is given by $\varphi_a(Z; \eta)h(X)$ (Lemma A.1 in the appendix).

Now we provide an influence-function-based semiparametric estimator for $\psi_a$. Following [8, 22, 43, 59], we propose to use *sample splitting* to allow for arbitrarily complex nuisance estimators $\widehat{\eta}$. Specifically, we split the data into $K$ disjoint groups, each with size of $n/K$ approximately, by drawing variables $(B_1, ..., B_n)$ independent of the data, with $B_i = b$ indicating that subject $i$ was split into group $b \in \{1, ..., K\}$. Then the semiparametric estimator for $\psi_a$ based on the efficient influence function and sample splitting is given by

$$\widehat{\psi}_a = \frac{1}{K} \sum_{b=1}^{K} \mathbb{P}_n^b \{\varphi_a(Z; \widehat{\eta}_{-b})h(X)\} \equiv \mathbb{P}_n \{\varphi_a(Z; \widehat{\eta}_{-B_K})h(X)\}, \tag{3}$$

where we let $\mathbb{P}_n^b$ denote empirical averages over the set of units $\{i : B_i = b\}$ in the group $b$ and let $\widehat{\eta}_{-b}$ denote the nuisance estimator constructed only using those units $\{i : B_i \neq b\}$. Under weak regularity conditions, this semiparametric estimator attains the efficiency bound with the double robustness property, and allows us to employ nonparametric machine learning methods while achieving the $\sqrt{n}$-rate of convergence and valid inference under weak conditions (see Lemma A.1 in the appendix for the formal statement). If one is willing to rely on appropriate empirical process conditions (e.g., Donsker-type or low entropy conditions [53]), then $\eta$ can be estimated on the same sample without sample splitting. However, this would limit the flexibility of the nuisance estimators.

The classification risk $\mathcal{L}(\beta)$ is a sum of two functionals, each of which is in the form of $\psi_a$, Thus, for each $\beta$, we propose to estimate the classification risk using (3) as follows

$$\widehat{\mathcal{L}}(\beta) = -\mathbb{P}_n \left\{ \varphi_a(Z; \widehat{\eta}_{-B_K}) \log \sigma(\beta^\top \boldsymbol{b}(V)) + (1 - \varphi_a(Z; \widehat{\eta}_{-B_K})) \log(1 - \sigma(\beta^\top \boldsymbol{b}(V))) \right\}. \tag{4}$$

Now that we have proposed the efficient method to estimate the counterfactual component $\mathcal{L}(\beta)$, in what follows we provide an approximating program for (P) which we aim to actually solve by substituting $\widehat{\mathcal{L}}(\beta)$ for $\mathcal{L}(\beta)$

$$\begin{aligned} \underset{\beta \in \mathcal{B}}{\text{minimize}} \quad & \widehat{\mathcal{L}}(\beta) \\ \text{subject to} \quad & \beta \in \mathcal{S}. \end{aligned} \tag{$\widehat{\mathsf{P}}$}$$

Let $\widehat{\beta} \in \mathsf{s}^*(\widehat{\mathsf{P}})$. Then $\widehat{\beta}$ is our estimator for $\beta^*$. We summarize our algorithm detailing how to compute the estimator $\widehat{\beta}$ in Algorithm 1.

($\widehat{\mathsf{P}}$) is a smooth nonlinear optimization problem whose objective function depends on data. Unfortunately, unlike (P), ($\widehat{\mathsf{P}}$) is not guaranteed to be convex in finite samples even if $\mathcal{S}$ is convex. Non-convex problems are usually more difficult than convex ones due to high variance and slow computing time. Nonetheless, substantial progress has been made recently [5, 42], and a number of efficient global optimization algorithms are available in open-source libraries (e.g., NLOPT). Also in order for more flexible implementation, one may adapt neural networks for our approach without the need for specifying $\sigma$ and $\boldsymbol{b}$; we discuss this in more detail in Section 6 as a promising future direction.

# 4 Asymptotic Analysis

This section is devoted to analyzing the rates of convergence and asymptotic distribution for the estimated optimal solution $\widehat{\beta}$. Unlike stochastic optimization, analysis of the statistical properties of optimal solutions to a general counterfactual optimization problem appears much more sparse. In what was perhaps the first study of the problem, [24] analyzed asymptotic behavior of optimal solutions for a particular class of nonlinear counterfactual optimization problems that can be cast into a parametric program with finite-dimensional stochastic parameters. However, the true program (P) does not belong to the class to which their analysis is applicable. Here, we derive the asymptotic properties of $\widehat{\beta}$ by considering similar assumptions as in [24].

We first introduce the following assumptions for our counterfactual component estimator $\widehat{\mathcal{L}}$.

(A1) $\mathbb{P}(\widehat{\pi}_a \in [\epsilon, 1 - \epsilon]) = 1$ for some $\epsilon > 0$

(A2) $\|\widehat{\mu}_a - \mu_a\|_{2,\mathbb{P}} = o_\mathbb{P}(1)$ or $\|\widehat{\pi}_a - \pi_a\|_{2,\mathbb{P}} = o_\mathbb{P}(1)$

(A3) $\|\widehat{\pi}_a - \pi_a\|_{2,\mathbb{P}}\|\widehat{\mu}_a - \mu_a\|_{2,\mathbb{P}} = o_\mathbb{P}(n^{-\frac{1}{2}})$

Assumptions (A1) - (A3) are commonly used in semiparametric estimation in the causal inference literature [20]. Next, for a feasible point $\bar{\beta} \in \mathcal{S}$ we define the active index set.

**Definition 4.1** (Active set). *For $\bar{\beta} \in \mathcal{S}$, we define the active index set $J_0$ by*

$$J_0(\bar{\beta}) = \{1 \leq j \leq m \mid g_j(\bar{\beta}) = 0\}.$$

Then we introduce the following technical condition on $g_j$.

(B1) For each $\beta^* \in \mathsf{s}^*(\mathsf{P})$,

$$d^\top \nabla_\beta^2 g_j(\beta^*)d \geq 0 \quad \forall d \in \{d \mid \nabla_\beta g_j(\beta^*) = 0, j \in J_0(\bar{\beta})\}.$$

Assumption (B1) holds, for example, if each $g_j$ is locally convex around $\beta^*$. In what follows, based on the result of [47], we characterize the rates of convergence for $\widehat{\beta}$ in terms of the nuisance estimation error under relatively weak conditions.

**Theorem 4.1** (Rate of Convergence). *Assume that (A1), (A2), and (B1), hold. Then*

$$\textit{dist}\left(\widehat{\beta}, \mathsf{s}^*(\mathsf{P})\right) = O_\mathbb{P}\left(\|\widehat{\pi}_a - \pi_a\|_{2,\mathbb{P}}\|\widehat{\mu}_a - \mu_a\|_{2,\mathbb{P}} + n^{-\frac{1}{2}}\right).$$

*Hence, if we further assume the nonparametric condition (A3), we obtain*

$$\textit{dist}\left(\widehat{\beta}, \mathsf{s}^*(\mathsf{P})\right) = O_\mathbb{P}\left(n^{-\frac{1}{2}}\right).$$

Theorem 4.1 indicates that double robustness is possible for our estimator, and thereby $\sqrt{n}$ rates are attainable even when each of the nuisance regression functions is estimated flexibly at much slower rates (e.g., $n^{-1/4}$ rates for each), with a wide variety of modern nonparametric tools. Since $\mathcal{L}$ is continuously differentiable with bounded derivative, the consistency of the optimal value naturally follows by the result of Theorem 4.1 and the continuous mapping theorem. More specifically, in the following corollary, we show that the same rates are attained for the optimal value under identical conditions.

**Corollary 4.1** (Rate of Convergence for Optimal Value). *Suppose (A1), (A2), (A3), (B1) hold and let $v^*$ and $\widehat{v}$ be the optimal values corresponding to $\beta^* \in \mathsf{s}^*(\mathsf{P})$ and $\widehat{\beta}$, respectively. Then we have*
$|\widehat{v} - v^*| = O_\mathbb{P}\left(\|\widehat{\pi}_a - \pi_a\|_{2,\mathbb{P}}\|\widehat{\mu}_a - \mu_a\|_{2,\mathbb{P}} + n^{-\frac{1}{2}}\right).$

In order to conduct statistical inference, it is also desirable to characterize the asymptotic distribution of $\widehat{\beta}$. This requires stronger assumptions and a more specialized analysis [47]. Asymptotic properties of optimal solutions in stochastic programming are typically studied based on the generalization of the delta method for directionally differentiable mappings [e.g., 48–50]. Asymptotic normality is of particular interest since without asymptotic normality, consistency of the bootstrap is no longer guaranteed for the solution estimators [12].

We start with additional definitions of some popular regularity conditions with respect to (P).

**Definition 4.2** (LICQ). *Linear independence constraint qualification (LICQ) is satisfied at $\bar{\beta} \in \mathcal{S}$ if the vectors $\nabla_\beta g_j(\bar{\beta})$, $j \in J_0(\bar{\beta})$ are linearly independent.*

**Definition 4.3** (SC). *Let $L(\beta, \gamma)$ be the Lagrangian. Strict Complementarity (SC) is satisfied at $\bar{\beta} \in \mathcal{S}$ if, with multipliers $\bar{\gamma}_j \geq 0$, $j \in J_0(\bar{\beta})$, the Karush-Kuhn-Tucker (KKT) condition*

$$\nabla_\beta L(\bar{\beta}, \bar{\gamma}) := \nabla_\beta \mathcal{L}(\bar{\beta}) + \sum_{j \in J_0(\bar{\beta})} \bar{\gamma}_j \nabla_\beta g_j(\bar{\beta}) = 0,$$

*is satisfied such that $\bar{\gamma}_j > 0, \forall j \in J_0(\bar{\beta})$.*

LICQ is arguably one of the most widely-used constraint qualifications that admit the first-order necessary conditions. SC means that if the $j$-th inequality constraint is active, then the corresponding dual variable is strictly positive, so exactly one of them is zero for each $1 \leq j \leq m$. SC is widely used in the optimization literature, particularly in the context of parametric optimization [e.g., 50, 51]. We further require uniqueness of the optimal solution in (P).

(B2) Program (P) has a unique optimal solution $\beta^*$ (i.e., $\mathsf{s}^*(\mathsf{P}) \equiv \{\beta^*\}$ is singleton).

Note that under (B2) if LICQ holds at $\beta^*$, then the corresponding multipliers are determined uniquely [56]. In the next theorem, we provide a closed-form expression for the asymptotic distribution of $\widehat{\beta}$.

**Theorem 4.2** (Asymptotic Distribution). *Assume that (A1) - (A3), (B1), and (B2) hold, and that LICQ and SC hold at $\beta^*$ with the corresponding multipliers $\gamma^*$. Then*

$$n^{-\frac{1}{2}} \left( \widehat{\beta} - \beta^* \right) = \begin{bmatrix} \nabla_\beta^2 L(\beta^*, \gamma^*) & \mathsf{B} \\ \mathsf{B}^\top & 0 \end{bmatrix}^{-1} \begin{bmatrix} \mathbf{1} \\ \mathbf{0} \end{bmatrix}^\top \Upsilon + o_\mathbb{P}(1)$$

*for some $k \times |J_0(\beta^*)|$ matrix $\mathsf{B}$ and random variable $\Upsilon$ such that*

$$\Upsilon \xrightarrow{d} N\left(0, var\left(\varphi_a(Z; \eta) h_1(V, \beta^*) + \{1 - \varphi_a(Z; \eta)\} h_0(V, \beta^*)\right)\right),$$

*where*

$$\mathsf{B} = \left[ \nabla_\beta g_j(\beta^*)^\top, \ j \in J_0(\beta^*) \right],$$

$$h_1(V, \beta) = \frac{1}{\log \sigma(\beta^\top \boldsymbol{b}(V))} \boldsymbol{b}(V) \sigma(\beta^\top \boldsymbol{b}(V)) \{1 - \sigma(\beta^\top \boldsymbol{b}(V))\},$$

$$h_0(V, \beta) = -\frac{1}{\log(1 - \sigma(\beta^\top \boldsymbol{b}(V)))} \boldsymbol{b}(V) \sigma(\beta^\top \boldsymbol{b}(V)) \{1 - \sigma(\beta^\top \boldsymbol{b}(V))\}.$$

The above theorem gives explicit conditions under which $\widehat{\beta}$ is $\sqrt{n}$-consistent and asymptotically normal. We harness the classical results of [48] that use an expansion of $\widehat{\beta}$ in terms of an auxiliary parametric program. To show asymptotic normality of $\widehat{\beta}$, linearity of the directional derivative of optimal solutions in the parametric program is required. We have accomplished this based on an appropriate form of the implicit function theorem [11]. This is in contrast to [33] that relied on the structure of the smooth, closed-form solution estimator that enables direct use of the delta method. Lastly, our results in this section can be extended to a more general constrained nonlinear optimization problem where the objective function involves counterfactuals (see Lemmas B.1, B.2 in the appendix).

## 5 Simulation and Case Study

### 5.1 Simulation

We explore the finite sample properties of our estimators in the simulated dataset where we aim to empirically demonstrate the double-robustness property described in Section 3. Our data generation process is as follows:

$$V \equiv X = (X_1, ..., X_6) \sim N(0, I),$$
$$\pi_a(X) = \text{expit}(-X_1 + 0.5X_2 - 0.25X_3 - 0.1X_4 + 0.05X_5 + 0.05X_6),$$
$$Y = A\mathbb{1}\{X_1 + 2X_2 - 2X_3 - X_4 + X_5 + \varepsilon > 0\} + (1 - A)\mathbb{1}\{X_1 + 2X_2 - 2X_3 - X_4 + X_6 + \varepsilon < 0\},$$
$$\varepsilon \sim N(0, 1).$$

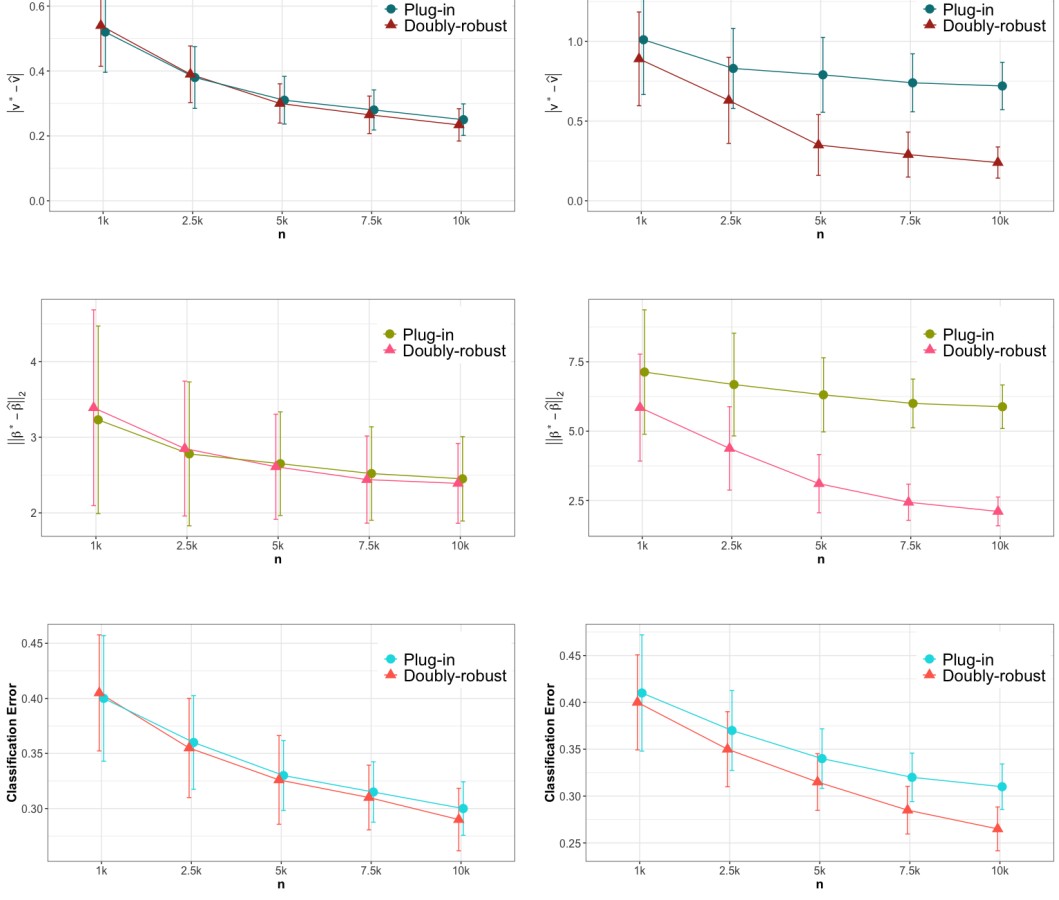

Figure 1: With correct $X$            Figure 2: With distorted $X$

Our classification target is $Y^1$. For $\boldsymbol{b}(X)$, we use $X$, $X^2$ and their pairwise products. We assume that we have box constraints for our solution: $|\beta_j^*| \leq 1$, $j = 1, ..., k$. Since there exist no other natural baselines, we compare our methods to the plug-in method where we use (2) for our approximating program $\widehat{\mathsf{P}}$. For nuisance estimation we use the cross-validation-based Super Learner ensemble via the SUPERLEARNER R package to combine generalized additive models, multivariate adaptive regression splines, and random forests. We use sample splitting as described in Algorithm 1 with $K = 2$ splits. We further consider two versions of each of our estimators, based on the correct and distorted $X$, where the distorted values are only used to estimate the outcome regression $\mu_a$. The distortion is caused by a transformation $X \mapsto (X_1 X_3 X_6, X_2^2, X_4/(1 + \exp(X_5)), \exp(X_5/2))$.

To solve $\widehat{\mathsf{P}}$, we first use the StoGo algorithm [40] via the NLOPTR R package as it has shown the best performance in terms of accuracy in the survey study of [35]. After running the StoGo, we then use the global optimum as a starting point for the BOBYQA local optimization algorithm [41] to further polish the optimum to a greater accuracy. We use sample sizes $n = 1k, 2.5k, 5k, 7.5k, 10k$ and repeat the simulation 100 times for each $n$. Then we compute the average of $|v^* - \widehat{v}|$ and $\|\beta^* - \widehat{\beta}\|_2$. Using the estimated counterfactual predictor, we also compute the classification error on an independent sample with the equal sample size. Standard error bars are presented around each point. The results with the correct and distorted $X$ are presented in Figures 1 and 2, respectively.

With the correct $X$, it appears that the proposed estimator performs as well or slightly better than the plug-in methods. However, in Figure 2 when $\widehat{\mu}_a$ is constructed based on the distorted $X$, the proposed estimator gives substantially smaller errors in general and improves better with $n$. This is indicative of the fact that the proposed estimator has the doubly-robust, second-order multiplicative bias, thus supporting our theoretical results in Section 4.

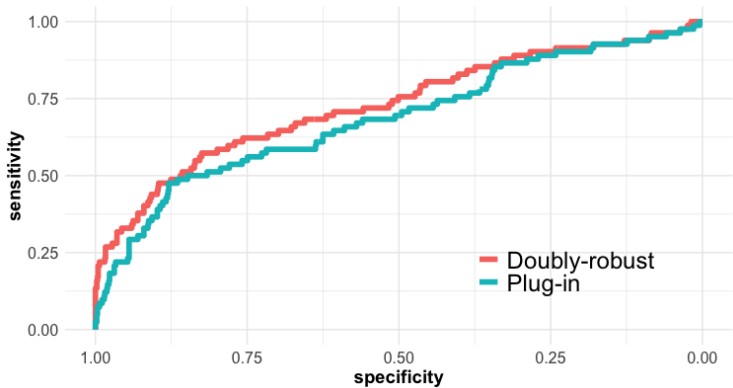

Figure 3: ROC curves

| Method | AUC | Accuracy |
|---|---|---|
| Plug-in | 0.692 | 0.64 |
| Doubly-Robust | **0.718** | **0.68** |
| Raw COMPAS Score | 0.688 | 0.65 |

Table 1: AUC and classification accuracy

## 5.2 Case Study: COMPAS Dataset

Next we apply our method for recidivism risk prediction using the Correctional Offender Management Profiling for Alternative Sanctions (COMPAS) dataset [2]. This dataset was originally designed to assess the COMPAS recidivism risk scores, and has been utilized for studying machine bias in the context of algorithmic fairness [2]. More recently, the dataset has been reanalyzed in the framework of counterfactual outcomes [32–34]. Here, we focus purely on predictive purpose. We let $A$ represent pretrial release, with $A = 0$ if defendants are released and $A = 1$ if they are incarcerated, following methodology suggested by [34].[3] We aim to classify the binary counterfactual outcome $Y^0$ that indicates whether a defendant is rearrested within two years, should the defendant be released pretrial. We use the dataset for two-year recidivism records with five covariates: age, sex, number of prior arrests, charge degree, and race. We consider three racial groups: Black, White, and Hispanic. We split the data ($n = 5787$) randomly into two groups: a training set with 3000 observations and a test set with the rest. Other model settings remain the same as our simulation in the previous subsection, including the box constraints.

Figure 3 and Table 1 show that the proposed doubly-robust method achieves moderately higher ROC AUC and classification accuracy than both the plug-in and the raw COMPAS risk scores. This comparative advantage is likely to increase in settings where we expect the identification and regularity assumptions to be more likely to hold, for example, where we can have access to more covariates or more information about the treatment mechanism.

## 6 Discussion

In this paper we studied the problem of counterfactual classification under arbitrary smooth constraints, and proposed a doubly-robust estimator which leverages nonparametric machine learning methods. Our theoretical framework is not limited to counterfactual classification and can be applied to other settings where the estimand is the optimal solution of a general smooth nonlinear programming problem with a counterfactual objective function; thus, we complement the results of [24, 33], each of which considered a particular class of smooth nonlinear programming.

---

[2] https://github.com/propublica/compas-analysis

[3] The dataset itself does not include information whether defendants were released pretrial, but it includes dates in and out of jail. So we set the treatment $A$ to 0 if defendants left jail within three days of being arrested, and 1 otherwise, as Florida state law generally requires individuals to be brought before a judge for a bail hearing within 2 days of arrest [34, Section 6.2].

We emphasize that one may use our proposed approach for other common problems in causal inference, e.g., estimation of the contrast effects or optimal treatment regimes, even under runtime confounding and/or other practical constraints. We may accomplish this by simply estimating each component $\mathbb{E}[Y^a \mid X]$ via solving (P) for different values of $a$, and then taking the conditional mean contrast of interest. We can also readily adapt our procedure (P) for such standard estimands, for example by replacing $Y^a$ with the desired contrast or utility formula, in which the influence function will be very similar to those already presented in our manuscript. In ongoing work, we develop extensions for estimating the CATE and optimal treatment regimes under fairness constraints.

Although not explored in this work, our estimation procedure could be improved by applying more sophisticated and flexible modeling techniques for solving (P). One promising approach is to build a neural network that minimizes the loss (4) with the nuisance estimates $\{\varphi_a(Z_i; \widehat{\eta}_{-B_K})\}_i$ constructed on the separate independent sample; in this case, $\beta$ is the weights of the network where $k \gg k'$. Importantly, in the neural network approach we do not need to specify and construct the score and basis functions; the ideal form of those unknown functions are learned through backpropagation. Hence, we can avoid explicitly formulating and solving a complex non-convex optimization problem. Further, one may employ a rich source of deep-learning tools. In future work, we plan to pursue this extension and apply our methods to a large-scale real-world dataset.

We conclude with other potential limitations of our methods, and ways in which our work could be generalized. First, we considered the fixed feasible set that consists of only deterministic constraints. However, sometimes it may be useful to consider the general case where $g_j$'s need to be estimated as well. This can be particularly helpful when incorporating general fairness constraints [14, 33, 34]. Dealing with the varying feasible set with general nonlinear constraints is a complicated task and requires even stronger assumptions [48]. As future work, we plan to generalize our framework to the case of a varying feasible set. Next, although we showed that the counterfactual objective function is estimated efficiently via $\widehat{\mathcal{L}}$, it is unclear whether the solution estimator $\widehat{\beta}$ is efficient too, due to the inherent complexity of the optimal solution mapping in the presence of constraints. We conjecture that one may show that the semiparametric efficiency bound can also be attained for $\widehat{\beta}$ possibly under slightly stronger regularity assumptions, but we leave this for future work.

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
