# APPENDIX

# A   Additional Technical Results

**Extra notations.** We let $\mathbb{B}_r(z)$ denote an open ball of radius $r$ centered at $z$, and let $\|M\|_F$ denote the Frobenius norm. $\|\cdot\|_2$ is understood as the spectral norm when it is used with a matrix. Further, for any vector-valued function $h : \mathbb{R}^{d_\theta} \to \mathbb{R}^l$ of arbitrary dimensionality $l$ whose first-order partial derivatives exist, we denote its Jacobian matrix with respect to a variable $\theta$ by $\boldsymbol{J}_\theta(h) \in \mathbb{R}^{l \times d_\theta}$.

Here we present additional notions and results which we will use for proofs.

**Definition A.1** (Quadratic growth condition)**.** *For each $\beta^* \in \mathsf{s}^*(\mathsf{P})$, there exists a neighborhood $\mathbb{B}_r(\beta^*)$ with some $r > 0$ and a positive constant $\kappa$ such that*

$$\mathcal{L}(\beta) \geq \mathcal{L}(\beta^*) + \kappa \boldsymbol{dist}(\beta, \mathsf{s}^*(\mathsf{P}))$$

*for all $\beta \in \mathbb{B}_r(\beta^*)$.*

The above quadratic growth condition is widely used in nonlinear programming and can be ensured by various forms of second order sufficient conditions [e.g., 51]. Next, we provide the following lemma that underpins the construction of our estimator in Section 3.

**Lemma A.1.** *For some fixed functions $g : \mathcal{Y} \to \mathbb{R}$ and $h : \mathcal{X} \to \mathbb{R}$, let $\mu_{g,a} = \mathbb{E}[g(Y) \mid X, A = a]$, so $\eta = \{\pi_a, \mu_{g,a}\}$. For any random variable $T$, let*

$$\varphi_a(T; \eta) = \frac{\mathbb{1}(A = a)}{\pi_a(X)} \{T - \mathbb{E}[T \mid X, A]\} + \mathbb{E}[T \mid X, A = a],$$

*denote the uncentered efficient influence function for the parameter $\mathbb{E}\{\mathbb{E}[T \mid X, A = a]\}$. Also, define our parameter and the corresponding estimator by $\psi_{g,a} = \mathbb{E}[g(Y^a)h(X)]$ and $\widehat{\psi}_{g,a} = \mathbb{P}_n\{\varphi_a(g(Y); \widehat{\eta})h(X)\}$, respectively. If we assume that:*

*(D1)  either i) $\widehat{\eta}$ are estimated using sample splitting or ii) the function class $\{\varphi_a(\cdot; \eta) : \eta \in (0,1)^2 \times \mathbb{R}^2\}$ is Donsker in $\eta$*

*(D2)  $\mathbb{P}(\widehat{\pi}_a \in [\epsilon, 1 - \epsilon]) = 1$ for some $\epsilon > 0$*

*(D3)  $\|\varphi_a(\cdot; \widehat{\eta}) - \varphi_a(\cdot; \eta)\|_{2,\mathbb{P}} = o_{\mathbb{P}}(1)$,*

*Then we have*

$$\|\widehat{\psi}_{g,a} - \psi_{g,a}\|_2 = O_{\mathbb{P}} \left( \|\widehat{\pi}_a - \pi_a\|_{2,\mathbb{P}} \|\widehat{\mu}_{g,a} - \mu_{g,a}\|_{2,\mathbb{P}} + n^{-1/2} \right).$$

*If we further assume that*

*(D4)  $\|\widehat{\psi}_{g,a} - \psi_{g,a}\|_{2,\mathbb{P}} \|\widehat{\mu}_{g,a} - \mu_{g,a}\|_{2,\mathbb{P}} = o_{\mathbb{P}}(n^{-1/2})$,*

*then*

$$\sqrt{n}(\widehat{\psi}_{g,a} - \psi_{g,a}) \xrightarrow{d} N\left(0, var\{\varphi_a(g(Y); \eta)h(X)\}\right), \tag{5}$$

*and the estimator $\widehat{\psi}_{g,a}$ achieves the semiparametric efficiency bound, meaning that there are no regular asymptotically linear estimators that are asymptotically unbiased and with smaller variance[4].*

*Proof.* The proof is indeed very similar to that of the conventional doubly robust estimator for the mean potential outcome, and we only give a brief sketch here.

Let us introduce an operator $\mathcal{IF} : \psi \to \varphi$ that maps functionals $\psi : \mathbb{P} \to \mathbb{R}$ to their influence functions $\varphi \in L_2(\mathbb{P})$. Then it suffices to show that $\mathcal{IF}(\psi_{g,a}) = \mathcal{IF}(\mathbb{E}[\mu_{g,a}(X)h(X)]) = \varphi_a(g(Y); \eta)h(X)$. In the derivation of the efficient influence function of the general regression

---

[4]This is also a local asymptotic minimax lower bound.

function in Section 3.4 of [23], when $h$ is known and only depends on $X$, it is clear to see that pathwise differentiability [23, Equation (6)] still holds when $h(x)$ is multiplied and thus

$$\mathcal{IF}(\mu_{g,a}(x)h(x)) = \frac{\mathbb{1}(X = x, A = a)}{\mathbb{P}(X = x, A = a)} \{g(Y)h(x) - \mu_{g,a}(x)h(x)\}$$
$$= \mathcal{IF}(\mu_{g,a}(X))h(X).$$

Hence, $\mathcal{IF}(\mathbb{E}[\mu_{g,a}(X)h(X)]) = \varphi_a(g(Y); \eta)h(X)$.

Another way to see this is that since the influence function is basically a (pathwise) derivative (i.e., Gateaux derivative) we can think of multiplying by $h(x)$ as multiplying by a constant, which does not change the form of the original derivative, beyond multiplying by the "constant" $h(x)$. We refer the reader to [23] and references therein for more details about the efficient influence function and influence function-based estimators. $\qquad\square$

## B  Proofs

For proofs, let us consider the following more general form of stochastic nonlinear programming with deterministic constraints and some finite-dimensional decision variable $x$ in some compact subset $\mathcal{S} \in \mathbb{R}^k$:

$$\begin{aligned} \underset{x \in \mathcal{S}}{\text{minimize}} \quad & f(x) \\ \text{subject to} \quad & g_j(x) \le 0, \quad j = 1, ..., m \end{aligned} \qquad (\mathsf{P}_{nl}) \qquad \begin{aligned} \underset{x \in \mathcal{S}}{\text{minimize}} \quad & \widehat{f}(x) \\ \text{subject to} \quad & g_j(x) \le 0, \quad j = 1, ..., m. \end{aligned} \qquad (\widehat{\mathsf{P}}_{nl})$$

We consider the case that $f, \widehat{f}$ are $C^1$ functions. In the proofs, the active set $J_0$ is defined with respect to $\mathsf{P}_{nl}$.

### B.1  Proof of Theorem 4.1

**Lemma B.1.** *Let $\widehat{x} \in \mathsf{s}^*\left(\widehat{\mathsf{P}}_{nl}\right)$ and assume that $f$ is twice differentiable with Hessian positive definite. Then under Assumption (B1) we have*

$$\mathsf{dist}\left(\widehat{x}, \mathsf{s}^*(\mathsf{P}_{nl})\right) = O\left(\sup_{x'} \|\nabla_x \widehat{f}(x') - \nabla_x f(x')\|\right).$$

*Proof.* Due to the positive definiteness of the Hessian of $f$, from the KKT condition at $x^* \in \mathsf{s}^*(\mathsf{P}_{nl})$ with multipliers $\gamma_j^*$

$$\nabla_x L(x^*, \gamma^*) = \nabla_x f(x^*) + \sum_{j \in J_0(x^*)} \gamma_j^* \nabla_x g_j(x^*) = 0,$$

it follows that the following second order condition holds:

$$d^\top \nabla_x^2 L(x^*, \gamma^*) d > 0 \quad \forall d.$$

Hence, by Still [51, Theorem 2.4] the quadratic growth condition holds at $x^*$. Then by Shapiro [47, Lemma 4.1] and the mean value theorem, we have

$$\mathsf{dist}\left(\widehat{x}, \mathsf{s}^*(\mathsf{P}_{nl})\right) \le \alpha \left(\sup_{x'} \|\nabla_x \widehat{f}(x') - \nabla_x f(x')\|\right)$$

for some constant $\alpha > 0$, which completes the proof. $\qquad\square$

Now, by the fact that both of the objective functions in (P) and $(\widehat{\mathsf{P}})$ are differentiable with respect to $\beta$, by Lemma A.1 and B.1, we obtain the result.

## B.2 Proof of Theorem 4.2

**Lemma B.2.** *Assume that $f$ is twice differentiable whose Hessian is positive definite. Then under Assumption (B1), (B2), if LICQ and SC hold at $x^*$, we have*

$$n^{1/2}\left(\widehat{x} - x^*\right) \xrightarrow{d} \begin{bmatrix} \nabla_x^2 f(x^*) + \sum_j \gamma_j^* \nabla_x^2 g_j(x^*) & \mathsf{B}(x^*) \\ \mathsf{B}^\top(x^*) & 0 \end{bmatrix}^{-1} \begin{bmatrix} \mathbf{1} \\ \mathbf{0} \end{bmatrix} \Upsilon,$$

*where*

$$n^{1/2}\left(\nabla_x \widehat{f}(x^*) - \nabla_x f(x^*)\right) \xrightarrow{d} \Upsilon.$$

*Proof.* First consider the following auxiliary parametric program with respect to $(\mathsf{P}_{nl})$ with the parameter vector $\xi \in \mathbb{R}^k$.

$$\begin{aligned} \underset{x \in \mathcal{S}}{\text{minimize}} \quad & f(x) + x^\top \xi \\ \text{subject to} \quad & g_j(x) \le 0, \quad j = 1, ..., m. \end{aligned} \tag{$\mathsf{P}_\xi$}$$

$(\mathsf{P}_\xi)$ can be viewed as a perturbed program of $(\mathsf{P}_{nl})$; for $\xi = 0$, $(\mathsf{P}_\xi)$ coincides with the program $(\mathsf{P}_{nl})$. Here, the parameter $\xi$ will play a role of medium that contain all relevant stochastic information in $(\widehat{\mathsf{P}}_{nl})$ [48]. Let $\bar{x}(\xi)$ denote the solution of the program $\mathsf{P}_\xi$. Clearly, we get $\bar{x}(0) = x^*$.

We have already shown that $\widehat{x} \xrightarrow{p} x^*$ at the rate of $n^{1/2}$ and that the quadratic growth condition holds at $x^*$ under the given conditions in Theorem 4.1. Further, since the Hessian $\nabla_x^2 f(x^*)$ is positive definite and LICQ holds at $x^*$, the uniform version of the quadratic growth condition also holds at $\bar{x}(\xi)$ (see Shapiro [48, Assumption A3]). Hence by Shapiro [48, Theorem 3.1], we get

$$\widehat{x} = \bar{x}(\xi) + o_\mathbb{P}(n^{-1/2})$$

where

$$\xi = \nabla_x \widehat{f}(x^*) - \nabla_x f(x^*).$$

If $\bar{x}(\xi)$ is Frechet differentiable at $\xi = 0$, we have

$$\bar{x}(\xi) - x^* = D_0\bar{x}(\xi) + o(\|\xi\|),$$

where the mapping $D_0\bar{x} : \mathbb{R}^k \to \mathbb{R}^k$ is the directional derivative of $\bar{x}(\cdot)$ at $\xi = 0$. Since $\bar{x}(0) = x^*$, this leads to

$$n^{1/2}\left(\widehat{x} - x^*\right) = D_0\bar{x}(n^{1/2}\xi) + o_\mathbb{P}(1).$$

Now we shall show that such mapping $D_0\bar{x}(\cdot)$ exists and is indeed linear. To this end, we will show that $\bar{x}(\xi)$ is locally totally differentiable at $\xi = 0$, followed by applying an appropriate form of the implicit function theorem. Define a vector-valued function $H \in \mathbb{R}^{(k+m)}$ by

$$H(x, \xi, \gamma) = \begin{pmatrix} \nabla_x f(x) + \sum_j \gamma_j \nabla_x g_j(x) + \xi \\ \mathrm{diag}(\gamma)(g(x)) \end{pmatrix}$$

where a vector $g$ is understood as a stacked version of $g_j's$. Due to the SC and LICQ conditions, the solution of $H(x, \xi, \gamma) = 0$ satisfies the KKT condition for $(\mathsf{P}_\xi)$: i.e., $H(\bar{x}(\xi), \xi, \bar{\gamma}(\xi)) = 0$ where $\bar{\gamma}(\xi)$ is the corresponding multipliers. Now by the classical implicit function theorem [e.g., 11, Theorem 1B.1] and the local stability result [51, Theorem 4.4], there always exists a neighborhood $\mathbb{B}_{\bar{r}}(0)$, for some $\bar{r} > 0$, of $\xi = 0$ such that $\bar{x}(\xi)$ and its total derivative exist for $\forall \xi \in \mathbb{B}_{\bar{r}}(0)$. In particular, the derivative at $\xi = 0$ is computed by

$$\nabla_\xi \bar{x}(0) = -\boldsymbol{J}_{x,\gamma} H(\bar{x}(0), 0, \bar{\gamma}(0))^{-1}\left[\boldsymbol{J}_\xi H(\bar{x}(0), 0, \bar{\gamma}(0))\right],$$

where in our case $\bar{x}(0) = x^*, \bar{\gamma}(0) = \gamma^*$, and thus

$$\boldsymbol{J}_{x,\gamma} H(\bar{x}(0), 0, \bar{\gamma}(0)) = \begin{bmatrix} \nabla_x^2 f(x^*) + \sum_j \gamma_j^* \nabla_x^2 g_j(x^*) & \mathsf{B}(x^*) \\ \mathsf{B}^\top(x^*) & 0 \end{bmatrix},$$

with $B = [\nabla_x g_j(x^*)^\top, j \in J_0(x^*)]$, and

$$\boldsymbol{J}_\xi H(\bar{x}(0), 0, \bar{\gamma}(0)) = \begin{bmatrix} \mathbf{1} \\ \mathbf{0} \end{bmatrix}.$$

Here the inverse of $\boldsymbol{J}_{x,\gamma} H(\bar{x}(0), 0, \bar{\gamma}(0))$ always exists (see Still [51, Ex 4.5]). Therefore we obtain that

$$D_0 \bar{x}(n^{1/2}\xi) = \begin{bmatrix} \nabla_x^2 f(x^*) + \sum_j \gamma_j^* \nabla_x^2 g_j(x^*) & B(x^*) \\ B^\top(x^*) & 0 \end{bmatrix}^{-1} \begin{bmatrix} \mathbf{1} \\ \mathbf{0} \end{bmatrix} n^{1/2}\xi.$$

Finally, if $n^{1/2}\xi \xrightarrow{d} \Upsilon$, by Slutsky's theorem it follows

$$n^{1/2}(\widehat{x} - x^*) \xrightarrow{d} \begin{bmatrix} \nabla_x^2 f(x^*) + \sum_j \gamma_j^* \nabla_x^2 g_j(x^*) & B(x^*) \\ B^\top(x^*) & 0 \end{bmatrix}^{-1} \begin{bmatrix} \mathbf{1} \\ \mathbf{0} \end{bmatrix} \Upsilon.$$

$\square$

Then, the desired result for Theorem 4.2 immediately follows by the fact that

$$\nabla_\beta \mathcal{L} = -\mathbb{E}\left\{ Y^a(Z; \eta) h_1(V, \beta) + (1 - Y^a) h_0(V, \beta) \right\}$$

where

$$h_1(V, \beta) = \frac{1}{\log \sigma(\beta^\top \boldsymbol{b}(V))} \boldsymbol{b}(V) \sigma(\beta^\top \boldsymbol{b}(V))\{1 - \sigma(\beta^\top \boldsymbol{b}(V))\},$$

$$h_0(V, \beta) = -\frac{1}{\log(1 - \sigma(\beta^\top \boldsymbol{b}(V)))} \boldsymbol{b}(V) \sigma(\beta^\top \boldsymbol{b}(V))\{1 - \sigma(\beta^\top \boldsymbol{b}(V))\},$$

followed by applying Lemma A.1.