# OpenReview forum: "Doubly Robust Counterfactual Classification"
_NeurIPS.cc/2022/Conference — NeurIPS 2022 Accept_

### Official Review · Reviewer_Pnnz · 2022-07-05

**Rating:** 6
**Confidence:** 2
**Soundness:** 3 good
**Presentation:** 4 excellent
**Contribution:** 3 good

**Summary:**

This paper proposes to solve counterfactual classification problems by solving a doubly robust estimator of the cross-entropy loss with smooth nonlinear programming, incorporating constraints. Theoretical analysis is provided in the asymptotic limit, including the rate of convergence of the solution to the set of optimal solutions, and asymptotic normality of the solution under the assumption that there is a unique solution.

**Questions:**

lines 110-112: sigma is a function from \mathcal{B}\times\mathcal{X}, so b(V) should be in \mathcal{X}, but on line 112, b(V) seems to be in R^k? I see that later, authors restrict attention to the simple case of \sigma(\beta^Tb(V)), which means that \beta and b(V) indeed has to be in the same space. But before this restriction takes place, wouldn't it be better to retain generality and let b(V) take place in \mathcal{X}?
Also, could the authors comment on why they restrict to finite (k-dimensional) parameter space / basis functions? This seems to rule out many useful cases; for example, the authors list the use of kernels as an example, but useful kernels such as the Gaussian kernel seem to be ruled out by this finite-dimensional restriction.

**Limitations:**

Limitations are discussed at the end of the paper, and in my opinion, sufficiently. The work is mostly of theoretical nature, and I do not deem it necessary to discuss potential negative societal impact of the work.

**Strengths And Weaknesses:**

Strengths

This is a well-written paper with sound and clear presentation of the mathematics. It makes clear the problem it is trying to solve, as well as the method it deploys to solve it.

The cross entropy loss function is widely-used, and the way authors extended it to the counterfactual setting, subject to constraints, seems natural and intuitive to me. The scope of contribution might be a bit limited, as it feels like a rather simple and straightforward marriage of very well-known and widely-used problem and technique, I still think the theoretical analysis and the algorithms are valuable to the community.

I do not work directly in the field of counterfactual prediction, where the accurate prediction of each counterfactual case is important rather than a comparison between counterfactuals (e.g. CATE), but the authors convince me that it is an important problem. Given this, and given how commonplace and intuitively important classification is, I find it surprising that no attempt has been made so far to tackle this problem.

Weaknesses

I do not like the fact that the proof of Lemma A.1 was omitted. I had a look through [21], to see if things really carried over seamlessly, but at least to me, things weren't so obvious. In particular, I would like to see worked out how a factor of h(X) makes no other difference than simply multiplying the influence function by the same factor, and why the authors choose to work with the uncentred influence function rather than the centred one. Perhaps it's obvious to people working in the field, but I have not seen such formulations in other papers.

---

> ### Author Response · Authors · 2022-08-02
> **Response to your questions and comments in Weaknesses**
>
> Thank you for your time and helpful comments. In the following, we address each of them.
>
> - **Proof of Lemma A.1.** The proof is indeed very similar to that of the conventional DR estimator for the mean potential outcome. The key point is that multiplying $h(X)$ does not substantially alter the existing proof techniques when $h$ is known and only depends on $X$ (however, this would not be the case if $h$ depended on the unknown distribution). This can be seen by reviewing the influence function calculation in Section 3.4 of the review paper Kennedy (2022), where it is shown that $IF(\mu(x)h(x)) = IF(\mu(x))h(x)$. Another way to see this is that since the influence function is basically a (pathwise) derivative (i.e., Gateaux derivative) you can think of multiplying by $h(x)$ as multiplying by a constant, which does not change the form of the original derivative, beyond multiplying by the constant $h(x)$. This logic has also has been applied in other papers without proofs, including Kennedy et al. (2020, Ann. Statist.) and Kennedy et al. (2020, arxiv.2102.12034). Nonetheless, following the Reviewer's suggestion, we include a concise formal proof for Lemma A.1. in the appendix of the revised manuscript.
> - **Centered vs. uncentered IF.** The choice between the centered IF $\varphi^*$ or uncentered IF $\varphi$ does not alter the computation of the estimator, nor the theoretical results. The centered one would solve an estimating equation to construct the estimator, i.e., solving
>     $$
>          \mathbb{P}_n[ \widehat\varphi^*(\widehat\psi) ]=0.
>     $$
>     However, when $\varphi^* = \varphi - \psi$ (i.e., the IF is linear in the parameter, as in our case) then using the centered IF is equivalent to using the one-step estimator based on the uncentered IF
>     $$
>      \widehat\psi = \mathbb{P}_n(\widehat\varphi).
>     $$
>     Thus, there is no difference.
> - **Finite parameter space in $\mathbb{R}^k$ and basis function.** First, please note that $k$ is not necessarily equal to $d_x = dim(X)$ as it represents the dimension of an expanded feature space through the basis function $b(\cdot)$, and $\beta$ is the corresponding coefficient. Also, please note that the use of $V \subseteq X$ is solely for addressing runtime confounding.  In practice, $k$ is typically much larger than $d_x$: i.e., we have 5 covariates but $k$ can be 20 if we use the 2nd-order power series. \
>  We use a fixed and finite $k$ for several reasons. First and foremost, a fixed and finite $k$ enables fast root-$n$ rates and tractable inference. Without these conditions, our asymptotic results would not hold. Further, the causal inference literature often uses nonparametric estimation with this projection approach, where estimands are projected onto a finite-dimensional parametric model space (see, e.g., Neugebauer \& van der Laan 2007, Semenova \& Chernozhukov 2021, Kennedy et al. 2021), in order to obtain better efficiency and interpretability. \
>     However, as pointed out by the reviewer, one may consider varying $k$ as a function of $n$ especially in some high-dimensional problems; i.e., when $d_x$ increases with $n$. In this case, $k$ grows to infinity and we can gain more flexibility in modeling. However, we may lose the fast $\sqrt{n}$ rates, and additional regularity or structural conditions would be required to guarantee feasibility of the problem. We interpret the Reviewer was drawing a connection between our approach and RKHS methods, possibly as one way to address this. But note that typical RKHS results such as the representer theorem cannot be directly applied to our setting, since our problem depends on unobserved counterfactuals and has flexible constraints; however, such an approach could be pursued in future work to address the case of varying $k$. \
>     Finally, with the basis expansion here, we enable general transformation techniques which yield a much richer feature space. In our case, the Gaussian kernel (or even a kernel based on another radial basis function $\phi(\Vert x - z \Vert)$ for some function $\phi$) can indeed be used for this transformation by fixing the pivotal points (or knots) $z$ beforehand, in the same spirit of B-splines.  \
>     We will clarify the aforementioned points in the revised manuscript.

---

> > ### Comment · Reviewer_Pnnz · 2022-08-04
> > **Thank you for your response.**
> >
> > I thank the authors for their thorough response to my queries, which have been sufficiently answered. Since they were minor comments in the first place, I keep my overall evaluation.

---

> > > ### Author Response · Authors · 2022-08-07
> > > **Thank you!**
> > >
> > > Thank you again for the very helpful discussion. By addressing your helpful questions and comments, we believe our manuscript will be much improved as a result.

---

### Official Review · Reviewer_v6k8 · 2022-07-08

**Rating:** 5
**Confidence:** 3
**Soundness:** 3 good
**Presentation:** 3 good
**Contribution:** 3 good

**Summary:**

This paper proposes an approximating program to the original so-called counterfactual classification program that can incorporate flexible constraints. The approximating program utilizes doubly robust estimators so that the optimal solution is $\sqrt{n}$-consistent as long as nuisance functions satisfy the stated assumptions.

**Questions:**

1. If I am right, I think your "counterfactuals" include both observed and unobserved outcomes. More precisely, I think it should be called "potential outcomes" in your setting. It is inaccurate if regard counterfactuals as potential outcomes.

2. The key is to estimate Y^a more accurately. Why can't we just use mu_a(X) or varphi_a(Z; eta) to estimate Y^a, instead, we should solve beta from P?

3. Assumptions A2 and A3 seem to limit the convergence rate for nuisance parameters. Your claim that "beta can attain sqrt{n} consistent even nuisance parameters have slower rates" seems not consistent with the arguments A2&A3. As I expect, nuisance parameters should converge at a rate of o(n^{-1/4}) if beta is sqrt{n} consistent.

4. The quality of beta depends on i) the estimation of nuisance parameters pi_a and mu_a, ii) the plug-in or doubly robust learner. You only consider a case of ii), but I am wondering what the results would be if pi_a is not correctly specified. In this case, will the proposed method be better than the simple plug-in?

**Limitations:**

In Section 6, the authors well discuss the limitation and potential extensions.

**Strengths And Weaknesses:**

Originality: 4. The whole strategy is similar to DR-learner, and the work is close to "Fair double ensemble learning for observable and counterfactual outcomes." But involving a flexible constraint is interesting, and the asymptotic property analysis for beta is solid.

Quality: 3. I think the experiments are not so rich. There are also some typos or unclear statements.

Clarity: 4. The logistics and mathematical proofs are clear. But it would be great if the authors can state the connection between the so-called counterfactual classification and general causal inference problems.

Significance: 3. I think this method does not bring significant improvement compared with simpler methods.

---

> ### Author Response · Authors · 2022-08-02
> **Response to your questions and comments in Quality and Significance**
>
> Thank you for your time and helpful comments. We address each of them below.
>
> - **Q1.** Following the causal inference literature, we used “counterfactual outcome” in the same sense as “potential outcome." The term “counterfactual” generally refers to what would have happened under alternative, contrary to fact, conditions. However, as the reviewer notes, there is a subtle but important difference between the terms “potential outcome” and “counterfactual outcome,” and we agree that our $Y^a$ should be defined as a potential outcome. Following your suggestion, we will fix this accordingly in the revised manuscript.
> - **Q2.** To be precise, our goal is to learn a mapping from $X$ to $Y^a$. If the runtime confounding issue were not a concern, one natural, simpler way to perform the mapping would be to estimate the regression function $\mu_a(X)$ and simply use it for prediction (plug-in approach). This approach would provide more flexibility to capture a complex structure between $Y^a$ and $X$. The downside of this approach would be failure to  achieve fast $\sqrt{n}$ rates and exposure to the curse of dimensionality in nonparametric modeling. However, the proposed projection approach achieves the $\sqrt{n}$ rates and the projection parameters are often more interpretable, although they may not properly reflect the correct model. Overall, each method has advantages and disadvantages, and here we focus on the less-settled projection approach. This trade-off is also discussed in Remark 3 in Kennedy et al. (2021). We will address this more clearly in the introduction of the revised text. \
>     Lastly, it is not very straightforward to use $\varphi_a(Z;\eta)$ for predicting $Y^a$; first, it depends not only on $X$ but also $A$ and $Y$; second, even if one could evaluate $\varphi_a(Z;\eta)$, it would yield a suboptimal prediction because the resulting MSE would be greater than that of $E[Y^a\mid X] = \mu_a(X)$.
> - **Q3.** It is correct that conditions (A2) and (A3) do allow us to estimate nuisance parameters at slower-than-$\sqrt{n}$ rates. As the Reviewer notes, one sufficient condition is that all the nuisance functions converge to their true values at a faster-than-$n^{1/4}$ rate in the $L_2(\mathbb{P})$ norm (e.g., estimators consistent at $n^{5/16}$ rates). Lowering the objective from $\sqrt{n}$ to $n^{1/4}$ for the nuisance estimator convergence rate facilitates the use of nonparametric methods with far more flexibility; for example, these rates are attainable under smoothness, sparsity, or other nonparametric structural constraints. This is in contrast to the plug-in regression estimator where the rates of convergence for estimating $\beta$ are essentially given by the rates at which $\mu_a$ is estimated (Lemma B.1). As a result, for these estimators $\sqrt{n}$ rates cannot be attained in general with nonparametric methods.
> - **Q4.** When $\pi_a$ is not correctly specified, $\Vert \widehat{\pi}_a - \pi_a \Vert_{2,P}$ can be viewed as a constant, and so by Theorem 4.1 the rate of convergence will essentially be the same as that of the plug-in estimator in general. There is a caveat, however, since the proposed doubly-robust estimator may show worse finite-sample performance than the plug-in estimator if the constant factor is too large. We will describe this point in the revised text.
> - **Quality.** Thank you also for this comment. We will make corrections and clarifying edits in the revised manuscript, particularly in Section 5 where we describe the simulations and case study. We will also define the term ``plug-in'' more clearly.
> - **Significance.** We showed empirically and theoretically that our proposed methods perform far better than the simple plug-in estimator when the outcome regression model is misspecified, which is a meaningful contribution in settings where correct specification is difficult to verify. Importantly, our method is also significant in other applications. Specifically, since Lemmas B.1 and B.2 are applicable to any optimization problem requiring estimation of general smooth objective functions, our framework provides a way to study a new class of estimands that are the solutions to a nonlinear optimization problem involving potential outcomes. In general, such estimands cannot be obtained in closed form as in the analysis of [31]. Moreover, one can still use our proposed approach for estimating standard contrast effects in causal inference (e.g., the CATE), even under runtime confounding and/or other practical constraints. One can accomplish this by simply estimating each component $E[Y^a \mid X]$ via solving (P) for each $a$, and then for example taking the conditional mean contrast of interest. One can also straightforwardly adapt our procedure (P) to find such contrast effects, for example by replacing $Y^a$ with the desired contrast formula, in which the influence function will be very similar to those already presented in our manuscript. We will address this more clear.

---

> > ### Comment · Reviewer_v6k8 · 2022-08-03
> > **Further concerns**
> >
> > I would like to thank your careful answers. Below I still have some concerns.
> >
> > Q2: I don't think the authors addressed my concerns.
> >
> > 1. Why $\varphi_{a}$ would yield a suboptimal prediction? If so, will it affect the quality of $\beta$ solved from Eqn. (4)? Because in Eqn. (4), $\varphi_{a}$ is an estimated quantity, and the solved $\beta$ should depend on the estimation quality of $\varphi_{a}$.
> >
> > 2. You claim that the plug-in estimator may fail to achieve $\sqrt{n}$ rates. But here is the case when $\hat{\mu}_a$ converges to $\mu_a$ at $\sqrt{n}$ rates that would make the plug-in estimator has $\sqrt{n}$ rates, which satisfies your assumption (A2-A3).
> >
> > 3. What is the connection with R-learner?
> >
> > Q4: Why can it be viewed as a constant? If so, is it right about my above concern Q2.2? I think in this case, the plug-in estimator is good enough and can rival the proposed estimator. Do you have any results to show this point?
> >
> > In summary, my concerns lie in the effectiveness between the plug-in, IPW, and the proposed DR estimator. Additionally, I am confused about the necessity of solving (4). It would be better if the paper elaborates more on the reason why we need to do so.

---

> > > ### Author Response · Authors · 2022-08-06
> > > **Response to your follow-up questions**
> > >
> > > Thank you again for your time and helpful comments which have helped us to improve our work. In the following, we address each of your comments.
> > >
> > > * We apologize our previous response was unclear and incomplete. You raise interesting questions, which touch on the following elements. \
> > >     (a) Please kindly recall that our goal is to predict $Y^a$ based on $X$, before $A$ and $Y$ are even observed. Since $\varphi_{a}$ is a function of $X$, as well as $A$ and $Y$, using $\varphi_{a}$ to directly predict $Y^a$ would not be feasible in the first place. \
> > >     (b) Even if it were feasible (assuming that we were to have access to all $X,A,$ and $Y$), prediction based on $\varphi_{a}$ is still generally suboptimal since
> > >     $$
> > >      E[  (\varphi_a(Z) - Y^a)^2] \geq  E [(\mu_a(X) - Y^a)^2] .
> > >     $$\
> > >     Hence, $\varphi_{a}$ gives greater MSE than $\mu_a$; in this sense, it would be a suboptimal (less efficient) prediction than $\mu_a(X)$.\
> > >     With (a) and (b) above, we have tried to address more thoroughly your question "Why can't we just use $\varphi_a$ to estimate $Y^a$. Now, we turn to your follow-up question "will the fact that (4) involves $\varphi_a$ affect the quality of the proposed estimator?"  \
> > >     (c) Please note that in (4), $\varphi_a$ is used to efficiently estimate the expected loss (1), not used to directly estimate $Y^a$, where we model the probability $P(Y^a=1)$ through the parametric score function $\sigma(\beta^\top {b}(V))$ still without assuming any parametric assumption about the true relationship between $Y^a$ and $X$. In this case, the expected loss (1) can be estimated efficiently as shown in Lemma A.1, and so does the parameter $\beta$. We will make this point more clear in Section 3.
> > > * Here, too, we should have been more clear and we apologize for this. The reviewer is correct that if $\mu_a$ is estimated at root-n rates, then the plug-in estimator is already efficient.  However, such fast rates for regression functions are only attainable under strong parametric model assumptions (e.g., that the regression function is known up to a finite-dimensional parameter). In contrast, in nonparametric models where the regression function is only assumed to live in some infinite-dimensional function space, minimax rates are strictly slower than root-n, so no estimator $\hat{\mu}_a$ would ever attain root-n rates. For example, the best attainable rate for estimating an $s$-smooth $d$-dimensional regression function is $n^{-1/(2+d/s)}$, which is always slower than root-n for $s<\infty$. We stress that we are considering exactly these flexible nonparametric settings where plug-in estimators would generally be deficient.
> > > * In regard to the comparison with R-learner, one major difference is that the goal of R-learner is to efficiently estimate the CATE possibly by exploiting some structure between the difference of the two outcome regression functions while ours is to efficiently estimate a single outcome regression function. Further, the R learner using a subset of covariates ($V$) as in our setting is not straightforward to implement because in the original work it is confined to using all covariates.
> > > * Thank you also for this question. We now realize we should have been more clear. By saying $\pi_a$ is (completely) misspecified, we meant it is inconsistent: i.e., $\Vert \hat{\pi}_a - \pi_a \Vert = O_P(1)$. Another way to see this is that both $\pi_a$ and $\hat{\pi}_a$ are bounded so that the $L_2$ error has to be bounded as well. When $\pi_a$ is consistently estimated by $\hat{\pi}_a$ the error goes to zero, so assuming that it is only bounded means $\hat{\pi}_a$ can be misspecified. \
> > >     And you are correct. When only $\pi_a$ is misspecified, while $\mu_a$ is correctly specified, estimation based on $\mu_a$ would be comparable to the proposed estimator. However, please note that in practice we typically have more information about the treatment mechanism than the outcome process, so it may be preferable to model $\pi_a$ instead of $\mu_a$, also to facilitate valid inference (Rosenbaum 1987). We will add this discussion into Section 4 of our text.
> > >
> > > Please do not hesitate to leave comments if our answer is still unclear, or if we missed something.

---

> > > > ### Comment · Reviewer_v6k8 · 2022-08-09
> > > > **After rebuttal**
> > > >
> > > > Thanks for your careful answers. Most of my concerns are addressed. But I think this paper still lacks more experimental discussions. Also, there remain some technical points to clarify. In general, this paper gives solid theoretical results. Therefore, I keep the original rating.

---

### Official Review · Reviewer_Q7fg · 2022-07-10

**Rating:** 7
**Confidence:** 4
**Soundness:** 4 excellent
**Presentation:** 3 good
**Contribution:** 3 good

**Summary:**

This paper proposes to perform counterfactual classification (inferring categorical/binary potential outcomes) by minimizing a cross-entropy objective with constraints. The objective is not directly computable (because of unobserved counterfactual outcomes), so the authors propose to use uncentered efficient influence functions (based on model propensity and model expected outcome functions) to compute the objective. The authors then show double robustness properties of their estimate, i.e. they show that distance between their solution $\hat{\beta}$ and the true solution set $s^*(P)$ is upper bounded by the “estimation quality” $||\hat{\pi}_a-\pi_a||$ and $||\hat{\mu}_a-\mu_a||$ of the propensity & outcome models, and the achieved objective value $\hat{v}$ is similarly close to the optimal $v^*$. Under additional assumptions (LICQ, SC, B2), they show that $\hat{\beta}$ is $\sqrt{n}$-consistent and asymptotically normal. They illustrate the stated theoretical properties with a toy example, and they also evaluate the classification accuracy of their method on the COMPAS recidivism prediction task.

**Questions:**

- Why do you make a point about runtime confounding if you are setting V==X in the toy experiment? Since you are making a point about how your method can deal with runtime confounding, I would suggest you either (a) showcase this or (b) remove the points about runtime confounding.
- On L110, $\sigma: \mathcal{B}\times \mathcal{X} \rightarrow (0,1)$ – should it be $\mathcal{X}$ here? Since we are using the features b(V) as input?
- How do you justify the distorted X? Can you provide intuition/an explanation for why the plug-in method "fails" here? Would be good to put this in the text too.


**Limitations:**

Yes, I think the authors adequately address limitations of their work & concretely outline next steps.


**Strengths And Weaknesses:**

I will first list the weaknesses of the paper, then its strengths, then give minor comments.

$\textbf{Weaknesses:}$

- I think my biggest concern about the paper is that it is difficult to understand why we should care about solving problem (P) (Section 2). Typically, for categorical outcomes Y, people assess quantities like Relative Risk (RR) or (Conditional) Average Treatment Effects (CATE/ATE). They then can make claims about how good their estimates are relative to those “ground-truth” quantities (CATE/RR), and can analyze e.g., consistency & convergence to CATE/RR. But here it’s unclear why $\beta^*$ is an important quantity, and why we should care about consistency & rates of convergence of an estimate $\hat{\beta}$ to the “ground-truth” quantity $\beta^*$.
- Overall, I think the introduction needs some work. Currently, it is too short and does not motivate the proposed solution well. This is related to my first point – ideally, the intro should make it crystal clear why the reader/the causal inference community should care about solving the objective (P). Some concrete comments:

-- L35 “Our approach allows investigators to flexibly incorporate various constraints into the models [...] to accommodate a wide range of practical constraints relevant to their classification tasks.” – would be good to give examples here. Same comment for L54 “possibly with various constraints on our model.” Eventually, you do give some examples of constraints (L66/67) but these should be moved earlier in the text.

-- L37 “This poses both theoretical…” – what is “this” here? Ambiguous.

-- In general, the intro does not help me/the reader understand why your solution is appropriate, and what is wrong with alternative proposals.

--It wouldn’t hurt to further expand on the differences between your work and [31], i.e. expanding on the points in L77-79, and clearly delineating the differences.

$\textbf{Strengths:}$

Now that I’ve said the unpleasant stuff, I’d like to talk about the strengths of the paper. Suppose that the reader is convinced that problem (P) is worth solving, and that $\beta^*$ is indeed a quantity of interest.

- Sections 2,3,4 are very well written, and it is mostly easy to follow the authors’ reasoning as they guide the reader to their solution.
- The theoretical results stated in section 4 are nice, and the authors use the toy experiment (Sec 5.1) effectively to showcase their theoretical results.

Minor/Misc.:
- Please make the y-axis the same in Figs 1 and 2 (i.e., 1 scale per row) so we can see the effect of X distortion.
- L158: “Here, we propose a more efficient estimator”. Respectfully, you did not propose this. Rather, you can say something like “we use…” and cite a reference for uncentered influence functions. This is a minor point, just a matter of language.
- Any practical guidance for readers on how to pick the basis functions $b(\cdot)$? How does the choice of basis function affect the solution to the problem?
- In problem (P), I would just write $\sigma(\beta^\intercal b(V))$ from the beginning, not $\sigma(\beta, b(V))$, this just makes it more difficult to understand (since you end up just using $\sigma(\beta^\intercal b(V))$ anyway).

---

> ### Author Response · Authors · 2022-08-02
> **Response to your questions and comments in Weaknesses**
>
> Thank you for your positive feedback, helpful comments, and time. In what follows, we address each of your comments.
>
> - **Motivation for solving $(P)$.** In typical causal inference problems, we perform the following two-step procedure: \
>   +Step 1. Prediction of some functional of the distribution of a potential outcome  (e.g., $E[Y^a \mid X]$). \
>   +Step 2. Contrast of those functionals under two (or more) hypothetical interventions (e.g., the CATE). \
> Step 1 pertains to counterfactual prediction. In Step 2, the contrast (or risk) between different interventions commonly serves as the basis for subsequent decision-making, as the Reviewer pointed out; i.e., it informs the choice of hypothetical intervention thought to lead to the most favorable outcome. \
> First, we would like to emphasize that one can still use our proposed approach for Step 2; i.e., for the contrast, even under runtime confounding and/or other practical constraints. One can accomplish this by simply estimating each component $E[Y^a \mid X]$ via solving (P) for $a=0,1$, and then for example taking the conditional mean contrast of interest. One can also straightforwardly adapt our procedure (P) to find the contrast effects (at least for the difference), for example by replacing $Y^a$ with the desired contrast formula, in which the influence function will be very similar to those already presented in our manuscript. \
> Note, however, that Step 1 is often useful to support decision-making on its own. There are settings where the contrast effects or relative risk are less interesting than understanding what may happen if a subject was given a certain intervention. This is especially useful in clinical research when predicting risk in relation to treatment started after baseline: e.g., what would happen if we could intervene on the availability of a heart transplant that is a post-baseline treatment [10]. Moreover, in the context of multi-valued treatments, it could be more useful to estimate each individual conditional mean potential outcome separately than to estimate all the possible combinations of relative effects.\
>     Both our proposed projection approach and the simpler plug-in approach have advantages and disadvantages, but here, we focus on the former. Please see our response to Q2 of the review v6k8 for detailed discussion.\
> For your specific comments:
>         -- As recommended, we will move the examples of constraints (L66/67) to earlier in the text. \
>         -- `This' (L37) refers to the general smooth nonlinear constrained optimization problem involving the counterfactual outcomes, where the closed-form solution does not exist. As mentioned at the end of Section 4, this is another key improvement from [31], as their result relies on the particular form of quadratic programming whose solution has a closed-form expression. We analyzed asymptotic properties of optimal solutions in a more general class of counterfactual optimization problems than [31]. As suggested, we will clarify this in the revised text.\
> -- As suggested, based on the above discussion we will expand our introduction so that the motivation of our work becomes more clear to the readers.
>
> - **Q1.** We appreciate this observation. Accordingly, we have redone the simulation in Section 5.1 with the following changes in model specification:
>     $$
>         V = (X_1, ... ,X_4),
>     $$
>     $$
>         \pi_a(X) = \text{expit}(-X_1 + 0.5X_2 - 0.25X_3 - 0.1X_4).
>     $$
>     The new results look almost the same, except that the performance gaps are now slightly larger than before; estimation of $\pi_a$ is still easy while estimation of $\mu_a$ is more difficult. As suggested, we will update the revised manuscript to reflect these new settings and results.
>
> - **Q2.** The Reviewer is correct; this refers to the expanded feature space of $V$. We will fix this in the revised manuscript.
> - **Q3.** Estimating the outcome regression function is more challenging when using the distorted (transformed) covariates to estimate $\mu_a$, as we are relying on the wrong information. This yields larger errors for the plug-in approach. This simulation setup verifies our theoretical findings in Section 4.
> - **Minor/Misc.**  Thank you for these helpful suggestions. Accordingly, we will address all four points in the revised text.

---

> > ### Comment · Reviewer_Q7fg · 2022-08-04
> > **Thank you!**
> >
> > Thank you for your thorough & timely response!
> >
> > I am satisfied with all your answers, only 1 question remains for me:
> >
> > Regarding your answer to "Motivation for solving (P)," I am still a little confused. I am familiar with the literature on treatment effect estimation, so I understand how doing Step 1, i.e., expected (conditional) potential outcome estimation properly can automatically give you Step 2 (contrast estimation e.g. CATE or RR). To rephrase my question: how can I be guaranteed that your method will do a good job of Step 1? Is it true that solving problem (P) can lead us to estimate the "true" expected potential outcome well? Maybe this is already obviously true to you & I am missing something.
> > I hope my question is more clear now, please let me know if you need more clarification.
> >
> > Thanks again

---

> > > ### Author Response · Authors · 2022-08-06
> > > **Response to your follow-up question**
> > >
> > > Thank you again for your time and insightful question.
> > >
> > > When we project the score function into a finite-dimensional parametric model space associated with $b(\cdot)$, our method guarantees that the proposed $\hat{\beta}$ converges to the optimal $\beta^*$ that minimizes the true classification risk for $Y^a$ (eqn. (1)) at fast $\sqrt{n}$ rates. Hence, as long as the projection is reasonable, i.e., if $b(\cdot)$ is flexible enough to reflect a fairly large model space that can accurately model the true functional relationship between $Y^a$ and $X$, our proposed methods can do a good job to estimate $Y^a$. The causal inference literature often uses nonparametric estimation with this projection approach, where estimands are projected onto a finite-dimensional parametric model space (see, e.g., Kennedy et al. 2021, Neugebauer \& van der Laan 2007, Semenova \& Chernozhukov 2021), in order to obtain greater efficiency and better interpretability.
> > >
> > > Please do not hesitate to let us know if our answer is still unclear, or if we missed something. We believe this discussion will be very helpful to improve our introduction in the revised text.

---

> > > > ### Comment · Reviewer_Q7fg · 2022-08-06
> > > > **Thank you!**
> > > >
> > > > Thank you for this, makes more sense now!
> > > > Would definitely be helpful to include your explanation in the intro/methods somewhere, I think it would clarify to the reader that you are (in the end) estimating a quantity of interest after all -- I feel that connecting the objective (P) back to something readers care about (e.g., expected conditional potential outcomes) & giving guarantees about it would motivate the approach better.
> > > >
> > > > This was my last unaddressed concern. I trust the authors will incorporate all my feedback above, & have raised my confidence score from 3 to 4.
> > > >
> > > > Out of curiosity: you mention in L125 that b(V) can be a neural net -- have you tried this? If so, does it work well in practice? Wouldn't it help with the distorted X case in your synthetic experiment?

---

> > > > > ### Author Response · Authors · 2022-08-07
> > > > > **Thank you!**
> > > > >
> > > > > Thank you for the very helpful discussion.
> > > > >
> > > > > * We agree that incorporating our previous discussion will definitely help the reader to better understand the motivation of our optimization problem $(P)$. As suggested, in Section 2, we will also be more clear about our estimand, the expected conditional potential outcomes projected onto some parametric model space, so the reader can easily find the connection between what they usually care about and the program (P). Moreover, this eventually will help the reader to value our technical contributions where we leverage tools in both stochastic optimization and semiparametric estimation.  Again, thank you for providing thoughtful comments.
> > > > > * Regarding your additional question on neural network approach, no we have not tried this yet. We feel that is a promising direction since we do not need to specify \& construct the basis functions $b(\cdot)$ while still can flexibly estimate the unknown score function without explicitly formulating a nonlinear optimization problem. In order for this, we can build our own neural network using (2) as the loss function with the estimated nuisance parameters; then $\beta$ will be weights of the network. For some people who are familiar with deep learning, this approach could be more straightforward and appealing. However, one should go through the optimal architecture, hyperparameter search, etc., to optimize performance of the network. We will elaborate this a bit further in the discussion section as well. We are indeed planning on trying this in future applied research with large datasets.
> > > > >
> > > > > By addressing your helpful questions and comments, we believe our manuscript will be much improved as a result.

---

### Official Review · Reviewer_8jj2 · 2022-07-11

**Rating:** 6
**Confidence:** 3
**Soundness:** 4 excellent
**Presentation:** 4 excellent
**Contribution:** 3 good

**Summary:**

This paper proposes a doubly robust estimator for counterfactual logistic regression with flexible constraints. The authors provide a novel algorithm along with theoretical guarantees and empirical evaluation.

**Questions:**

See section above.

**Limitations:**

The authors have adequately addressed and potential negative societal impact of their work.

**Strengths And Weaknesses:**

**Strengths:**
* addresses a common problem in causal inference (robust counterfactual or effect estimation with binary outcome)
* provides theoretical guarantees for both estimation and inference
* the presentation is clear and cohesive
* the empirical study is compelling

**Weaknesses:**
* The title is somewhat misleading as it only tackles a specific type of parametric classification (logistic regression). Does not easily extend to more complex classification methods
* I wonder how practical the main algorithm is. While I appreciate that the constraints can be very general, I wager that most applications will require a $L_1/L_2$ constraint (penalty) on $\beta$. One advantage of the naive/natural estimator in (2) is that it is equivalent to a weighted logistic regression (because $\\hat{\\mu}_a(x)\in[0, 1]$) so it can be used with any black-box logistics regression implementation (e.g. in Python). This is not true for the proposed estimator since some of the weights will be negative. Is there any simplification of the algorithm in the case of $L_1/L_2$ constraints?

---

> ### Author Response · Authors · 2022-08-02
> **Response to your comments in Weaknesses**
>
> Thank you for your time and the helpful comments. We address each of them below.
>
> -  In the theoretical analyses and empirical demonstrations, we primarily focused on the logit model with basis expansion as our ``best" parametric model that approximates the true model. However, our work is considerably more general for the following reasons. First, as noted in Section 2, our framework and theoretical results can be extended from logistic regression to other techniques, as long as they use a smooth, finite-dimensional score function $\sigma$ that satisfies the required regularity assumptions. Theoretically one may also use neural networks with the proposed loss function and smooth activation functions.
>     As the Reviewer notes, while this extension is not practically straightforward, it could be explored in future work. Second, unlike in ordinary logistic regression, we do not use MLE for model fitting since we do not assume a `log-linear' true relationship. Following your suggestion, we will clarify this point in the revised manuscript. For these reasons, we are inclined to make these clarifications and keep the current title; however we are open to consider alternative titles if this seems insufficient.
>
> - This is a very helpful comment. $L_p$ constraints can be incorporated into our framework, but as the reviewer notes, it may not be feasible with off-the-shelf models in Python or $\texttt{R}$ (unless packages exist to effectively reflect negative weights); rather, we must formulate and solve the new optimization problem each time. While not the most convenient, this approach facilitates the nice asymptotic properties that we derived. To enhance practicality, we plan to disseminate a flexible template of the code to execute our methods with various constraints customizable by the user. As suggested, we will elaborate in the revised manuscript.

---

> > ### Comment · Reviewer_8jj2 · 2022-08-08
> > **Rebuttal Response**
> >
> > I thank the authors for the detailed response. I stand by my original (overall positive) assessment of this work.

---

> > > ### Author Response · Authors · 2022-08-08
> > > **Thank you!**
> > >
> > > Thank you again for your time and insightful comments! We believe this discussion will be very helpful to improve our revised text.

---

### Meta-Review · Area_Chair_ryVm · 2022-08-28

**Recommendation:** Accept
**Confidence:** Certain

**Metareview:**

Reviewers agreed that the paper makes interesting theoretical contributions to the problem of counterfactual classification, with a novel and valuable result. The reviewers pointed out several aspects in which the empirical evaluation could be more thorough, though it is my view that is definitely sufficient for validating the paper's main claims. Finally, the paper is well-written and presents the math crisply.

**Award:**

No

---

### Decision · Program_Chairs · 2022-09-14

Accept